# 'Nature Is Something We Can't Replace': Mexican Students' Views of the Landscape They Want to Conserve

**Eija Yli-Panula [1,](https://orcid.org) , Eila Jeronen [2] and Gabriela Rodriguez-Aflecht [1]**

[1] Department of Teacher Education, University of Turku, 20014 Turku, Finland; gabriela.rodrigues@utu.fi
[2] Faculty of Education, University of Oulu, 90014 Oulu, Finland; eila.jeronen@oulu.fi
* Correspondence: eija.yli-panula@utu.fi

**Abstract:** The primary aim of this qualitative study was to identify the landscapes that 7–12-year old Mexican students (n = 440) would like to conserve by analysing their drawings. Another aim was to determine the environmental relationship and environmental values of 5th and 6th graders (n = 152) by studying their texts. The data were analysed using mixed methods. In this study, landscape is understood as a visual experience of the environment, comprising the visible features of an area. Based on the results, all of the three main landscapes—nature, social and built—were deemed worth conserving. Beyond students' immediate environment, the polar regions, North America, Australia and Africa were mentioned; Europe and Asia were not. The landscape drawings were realistic and carefully made, and the descriptions attached to them were clearly written. The environmental approach was mainly humanistic, and aesthetic values were appreciated by both genders. Utilitarian values were mentioned more often by boys than girls. The students' descriptions reflected their environmental relationship, e.g., concern about nature, showing causal relationships, appreciation and affection. Concern or worry was often accompanied by the mention of human's responsibility in the students' texts, but they seldom considered their own activities in relation to the environment. The students depicted threats to nature, but they externalized themselves from the mechanisms threatening nature. In addition, they did not show familiarity with natural processes and scientific terminology. The study reveals that it is not only theoretically important to have distinct values, but these also need to be recognized by individuals. If the humans' pro-environmental actions are to be promoted through education, it is important to study students' values, as they may be important barriers to behavioral change. As students showed concern about preserving nature, teachers can discuss environmental values and different ways to take action and make changes with them, in order to avoid anxiety.

**Keywords:** environmental approach; environmental relationship; environmental values; landscape drawings and texts; qualitative study

## 1. Introduction

The knowledge of students' subjective worldviews [1], their environmental conceptions and environmental relationships, are crucial to strengthening children's connection to nature and expanding their knowledge base [2]. The present qualitative study aimed to explore Mexican (Monterrey) students' views of landscapes they deem worth conserving and to determine their environmental relationship and environmental values.

This study is part of a bigger project aiming to investigate what children and adolescents draw in their landscapes and deem worthy of conserving. The project has been carried out in several

countries. Mexico is an interesting place, due to its large diversity. Monterrey itself is an affluent, constantly growing city, close to the border with Texas, where children are very much exposed to United States culture.

The basis for this study lies in the word landscape. The concept of landscape refers to a visual experience. The Finnish word for landscape 'maisema' has a strong connection to the 'bounded area' or 'region', just like the German word 'Landschaft'. However, it differs from the English word 'landscape' which covers more social aspects [3,4] and is used in Monterrey school due to the school language of the participants of the study.

Childhood environments in connection to place attachment, identity and dependence, as well as cultural and environmental experiences, form the basis for the environmental relationship [5,6]. Environmental relationship depicts a person's attitude toward his or her environment [7], whereas environmental values describe different ways to approach the environment [8,9]. Values are something that people consider important to them. Kellert [9] has empirically refined a set of nine environmental values (utilitarian, naturalistic, ecologistic-scientific, aesthetic, symbolic, dominionistic, humanistic, moralistic, negativistic) which reflect what kind of values man associates with nature and the environment in general. According to Steg et al. [10], the distinction between the different types of values is not only theoretically meaningful but also recognized by individuals. So, if you want to promote pro-environmental actions through education, it is also good to study the values of the students, as these may be important barriers to behavioral change.

Many studies have shown that repeated nature experiences (e.g., [11,12]) and long-term nature education (e.g., [13]) develop environmental attitudes and views. Research shows that fewer children play outdoors and outdoor play is increasingly centered on the home, rather than the countryside, parks and beaches [14–16], because media and information technology take up more and more of youth's spare time [17]. Thus, schools and parents have a significant responsibility to provide children with experiences in nature and to develop a positive environmental relationship in children.

The qualitative data of the study consists of Mexican students' drawings and written descriptions of landscapes worth conserving. Drawings have been successfully used in previous environmental studies [18,19]. The use of drawings as data has been justified because children like to draw, and it is an easy and quick way to get information despite any language barriers. Previously, drawings of landscapes worthy of conservation have been collected from several countries, including Finland, Russia, Sweden and Nepal. Eloranta's [20] study explored landscape drawings of Finnish and Russian children and young people. According to the results, the girls drew more ecological landscapes, while the boys drew built and social landscapes. The girls also drew more animals than boys. In other countries, children have seen all three types of landscapes (nature, social and built landscapes) as worthy of conservation [20–23]. The results from Nepal, however, were different because they did not contain any nature landscapes [22]. Children especially appreciated a landscape where there were no signs or people of human activity, and they thought that man was not part of the natural world [20–24]. In the Yli-Panula and Eloranta [21] study, Finnish children and adolescents drew only a few animals in a landscape deemed worthy of conservation. Reasons for this may be that animals are difficult to draw and/or that animals move from place to place in their surroundings and are therefore not placed in a particular location. Moreover, according to previous studies, children and adolescents were worried about their environment. They wanted to conserve pure nature, so they did not include man in the landscape [15,21,23,25].

The first aim of this study was to identify what type of landscape 7–12-year-old Mexican students would like to conserve. The second aim was to determine the environmental relationship and the environmental values of the 5th and 6th graders. Using thematic analysis [26], the drawings and the texts of the students were divided into three groups: nature, social or built environments [27]. The Mexican students' environmental relationship was studied based on environmental relations and the ethical standards of nature [2,28,29]. Their environmental values were analysed by using Kellert's [8,9,30] categorization of environmental values. The results are presented for the students'

grade and gender. The findings are discussed in connection to landscape theories, environmental relationship theories and earlier findings of drawn and written descriptions of environments.

## 2. Theoretical Background

### 2.1. Core Concepts of the Study

The core concepts of this study are landscape, environment, environmental relationship and environmental values. There are various interpretations of the term 'landscape'. In geography, landscape refers, for example, to an area containing a mosaic of landscape elements and can be experienced [31,32]. 'Environment' can also mean all issues associated with human beings, including physical and social dimensions [27]. In this study, landscape is defined as thematic content, so it can be understood as a visual experience of the environment [33].

The concept of 'environmental relationship' describes a person's attitude toward his or her environment [6]. The development of a positive environmental relationship requires active work and contact with nature [17,27]. The interaction between man and environment is reciprocal. Not only does man affect the environment, but man is also affected by the environment [34]. Changes in the environment affect a person's experience and activities [27]. Nature experiences play an important role in the development of the environmental relationship.

There are many formulations of the essential components of environmental values [35,36]. Such formulations include, for example, living harmoniously within ecological systems, developing a caring, responsible attitude toward nature, and promoting a sense of continuity and community with other people and all living things. In this study, environmental values are related to those views which depict the underlying values of Mexican students in relation to the environment, and the role of humans and other living things in achieving sustainability.

### 2.2. Dimensions of Landscape and Environment

The landscape and the environment gain significance through individual experiences, and observations [6,27] have shown how environmental relations and images in space and place are created, and how they are shaped by knowledge, experience and thinking. Feelings, attitudes and a sense of worthiness also have an important role in structuring the environmental relationship and environmental images. The landscape and environment always have cultural implications, and thus can be seen as a part of the process by which the natural environment becomes a cultural landscape due to human activity [32].

According to Aura et al. [23], the environment can be divided into the natural environment, the built environment and the social environment. The natural environment includes all living and non-living things that occur naturally in a particular region [37]. The built environment includes all places built by humans, such as yards, parks, streets or channels [38]. Social environments encompass the immediate physical surroundings, social relationships and cultural milieus within which defined groups of people function and interact [39]. Together, the natural environment and the built environment form the physical environment, while the social environment includes people and different communities [23]. According to the previous classifications, the landscape is divided in this study into the nature, built and social landscapes. The nature landscape refers to an environment that has not been influenced by human activity for a long time. The built environment is synonymous with the built landscape, meaning the landscape in which man is present. The social environment is used as a synonym for the social landscape.

### 2.3. Views of People about Environment and Environmental Relationship

Many researchers have studied people's views of the environment and their environmental relationship. According to Loughland et al. [2], children and adolescents aged 9–17 described the environment in six different ways. In the first, and most limited view, the environment was seen as an

object only, and the students did not attach personal meanings to the environment. In the second, the environment was perceived as a place that contains life without man. In the third, man was taken into account as a part of the environment, but his role was not explained in any detail. In the fourth, the environment was thought of as a complex entity, which was perceived to make or give something to man. For example, the environment was perceived as a place of relaxation. In the fifth, man was a part of the environment and also responsible for it. The sixth emphasized the continuous interaction between man and the environment. The students understood that this interaction is mutual [2]. The last three views show that the students had a diverse understanding of the environment and that they were aware of their role as part of their environment. Such views contribute to the creation of environmentally friendly attitudes. The students' descriptions of environment also described their environmental relationship: how they felt about their environment and how they experienced it.

Findings similar to those of Lougland et al.'s [2] study have also been obtained in other environmental studies [15,40]. Children were found to be aware of global environmental problems and the need for environmental protection. They were also able to offer direct and indirect ways of taking care of the environment. In addition, children highlighted the importance of the environment as a place for recreation and relaxation, and they felt that the environment is important for human well-being [15].

Littunen and Lähde [28] distinguished four different levels of environmental relations and the ethical consideration of nature. The environmental relationship can exist at an individual level, whereby a person has a relationship with a single natural entity, such as her/his pet. At the next, and at a more interdimensional level, the environmental relationship is formed between man and various species. This may include, for example, the idea of protecting endangered species. On the third level, the protection of species is related to the protection of whole habitats. The most interdependent relationship is at the global level, for example, dealing with the problem of global warming [24]. In previous environmental studies, the children have been worried about the loss of proper living conditions for animals [18,19,35]. In Barraza's [18] study, nearly 40% of the students described environmental problems in their drawings, and half of the students thought that the state of the world would become worse in the next 50 years. The students' drawings included global environmental problems, such as pollution, acid rain, deforestation and lack of fresh water.

According to Willamo [29], there are three views of the human relationship with nature. In the first, nature is seen far from itself. The relationship with nature is remote, and nature is seen as being outside of man. In the second, man is seen as both part of and separate from nature, and two dimensions are distinguished in both man and nature: the human and the ecological one. In the third, man is considered to be one with nature.

## 2.4. Nature Experiences Play an Important Role in the Environmental Relationship

The environmental relationship is shaped by various factors, and the basis of the environmental relationship is created in childhood [5]. The home plays a major role in the formation of the environmental relationship. Environmentally friendly attitudes develop in children who are offered opportunities to observe and experience environmental issues, such as recycling, at home (Musser) [41]. Besides the home, schools and the media play important roles in developing environmental knowledge, attitudes and behaviour. For example, according to Lewis, Mansfield and Baudains [42], during and after completion of their environmental projects, the students were able to verbalise their environmental knowledge, explain associated values, express their attitudes toward local environmental issues and outline their behavioural intentions and actions to improve their environment. The media are effective in calling attention to environmental problems and making the public aware of the problems created by pollution, and, hence, they can be instrumental in creating public pressure to do something about them. Together with developing technologies, the media's importance and efficacy regarding environmental education has grown. However, since the media-mediated picture of the environment does not always meet the educational goals, students should be taught about media criticism [43].

According to Kellert [8], the concept of 'direct nature experience' means immediate physical contact with the natural environment, habitats, plants or animals. Direct nature experience is often spontaneous and unplanned. Today, true contacts with nature are often replaced by virtual channels providing children with transmitted nature experiences through photos, videos, images and metaphors. Therefore, children have fewer nature experiences than those of previous generations [14]. Their nature experiences are also increasingly organized and controlled by adults [8]. Consequently, nature can be something distant or exotic for the child.

Direct nature experiences play a crucial role in the development of the environmental relationship of children. They have an impact on children's affective and cognitive development [8]. The natural environment has a positive effect on the happiness and well-being of students [44]. It can also develop children's emotional affinity toward nature [8] and give rise to the appreciation of familiar and pleasant landscapes [17]. It can increase their positive mental, emotional and social health outcomes, such as their sense of achievement, self-confidence, self-esteem, adaptation to different learning styles, sensory engagement, skills in caring and nurturing, connectedness to others, feelings of freedom and creativity, and feelings of stress relief and engagement in school [45]. Also, Wistoft [46] has reported possible benefits on aspects of students' learning motivation. However, children's experiences of nature can be positive or negative, and both positive and negative feelings towards nature contribute to their affective development.

The environment shapes the child and the child's actions. Interactions may vary from social relations, to the triggering of senses and emotions, to concrete acts. Rickinson et al. [47] highlighted the benefits of the school grounds/community projects on students' science process skills, as well as the impact of fieldwork and visits on students' long-term memory and higher order learning. Active learning develops the learning skills of inquiry, experimentation, feedback, reflection, review and cooperative learning [48]. While students learn outdoor skills, they also learn things about their environment, themselves and each other. A balanced alternation of direct, indirect and transmitted nature experiences develops thinking skills through detection, research and reasoning [8]. According to Moeed and Averill [49], students are also able to transfer the gained skills into a different context. All of these may lead to greater knowledge of the subjects and the environment and to the development of the environmental relationship.

*2.5. Values behind the Environmental Relationship*

This study also explored the environmental values underlying the environmental relationship by evaluating the Mexican students' writings that were linked to their landscape drawings. Value is a multidisciplinary concept. According to Niiniluoto [50], there are instrumental values and intrinsic values. Instrumental values are useful as a means to achieve a goal that is considered valuable, while intrinsic values are important due to their essential nature. Rokeach [51] notes that values have cognitive, affective and functional factors associated with them.

Environmental values are formed during early and middle childhood, and their development requires interaction with the environment [52]. On the basis of empirical studies, Kellert [30] developed a nine-level environmental rating system that can be used to examine the human relationship with the environment.

Kellert [8,9,30] described the development of environmental values through three phases. The first stage occurs before six years of age or during early childhood. During this phase, utilitarian, dominant and negativistic environmental values develop. The utilitarian environmental value emphasizes materialistic benefit. Nature is used to satisfy man's own needs. The dominant environmental value means that, due to his own needs, man emphasizes his own existence at the expense of nature. Nature is thus considered subordinate to man. A negativist environmental value is related to a hostile or fearful attitude toward either an animal or a natural phenomenon.

The second stage is between six and twelve years of age when humanistic, symbolic and aesthetic environmental values develop. With the humanistic environmental value, human responsibility for

nature is recognized. The symbolic environmental value stresses that nature has a symbolic value. At the heart of aesthetic environmental value is the appreciation of beauty and seeing it in nature.

The third stage in the development of environmental values is from the age of thirteen to seventeen. At this stage, values are directed from local to global, and the stage is characterized by naturalistic, ecological–scientific and moral environmental values [8]. The naturalistic environmental value includes the appreciation of nature for the sake of nature itself. The ecological–scientific environmental value emphasizes understanding natural mechanisms and taking natural patterns of action into account. Dependency relationships are part of nature's activity; the link between the conservation of species and the protection of habitats is important. The moral environmental value stresses that everyone should be treated as well as possible and that damage to others should be minimized. Nature should be protected and cherished. The primary point of view is the appreciation of natural order and harmony.

Eagles and Muffit [53] showed in their study that the most common environmental values among 12 to 14-year olds are the humanistic, moral, naturalistic and ecological environmental values. Girls show more moral value than boys, while no gender differences are observed in the other environmental values. According to Eagles and Demare [5], moral and ecological environmental values are most prevalent among children who discuss the environment at home, view nature documents and read environmental literature.

## 3. Study Design

### 3.1. Study Questions

The first aim of this study was to find out what kind of landscapes the Mexican students want to conserve. The hypothesis was that the landscape which the students wanted to conserve would be important and valuable to them. The second aim was to determine what kind of environmental relationship and environmental values the students had.

The study questions were the following:

1. What kind of landscapes do Mexican students (1st–6th graders) want to conserve?
2. What kind of environmental values are revealed by Mexican students (5th–6th graders), and what do those values tell about their environmental relationship?

The first study question was related to nature, built and social landscapes as defined in environmental psychology [23], and the data included the Mexican students' (n = 40) drawings and accompanying texts. The second study question was related to the 5th–6th graders' (n = 152) landscape drawings with the accompanying texts. The variations in the content of the drawings were studied according to age and gender. The data were analysed and categorized using mixed methods.

### 3.2. Participants and Material of the Study

The participants were 1st–6th graders (N = 655) of an immersive English language private school in the city of Monterrey, Mexico. Mexico is one of the most diverse countries, culturally and economically, in Latin-America. The age of the students ranged from 7 to 12 years. Most spoke Spanish as a mother tongue, although the language of school was English, and were of high socioeconomic status. These background issues support students´ opportunities to travel abroad. The school was chosen based on collaboration connections and voluntariness. Data was collected by a Spanish–English bilingual postgraduate student who passed detailed instructions on data collection procedures to the principal. The school's elementary school principal was contacted by email in 2010, through which the aims of the study and the data collection procedures were explained. The principal then communicated with the teachers, who participated voluntarily. Teachers asked their students to draw on an A4 white sheet 'a landscape you would want to conserve' and include a short text describing the landscape in the drawing. Pencils, including colored pencils, were used. The term conserve was chosen because it includes aspects of the future, and if we do not take action to preserve the environment,

it will disappear. Students were given one class period lasting 45 min to work on their drawings. Demographic information (name, age, gender, grade level and name of the school) of the students was also collected and students were asked to write them in English.

The first research question focused on the landscapes the Mexican students (N = 440 1st–6th graders' drawings; 49% girls and 51% boys) wanted to conserve. The students' texts supported the interpretation of the drawings. The 5th–6th graders' (n = 152) drawings and texts were chosen for further analyses to find out the answer to the second research question about students' environmental values and their relationship with the environment.

### 3.3. Drawings as Study Material

Drawings have proven useful for studying children's environmental values, environmental relationships and environmental concepts [18,19]. Some advantages of this research method are that it avoids language problems and that children often feel comfortable drawing. Drawings are of special value for students that have difficulties expressing themselves verbally [54], and they can be helpful for those students who are shy or lack language skills [55]. When interpreting drawings, it must be taken into account that there are differences in respect to age and gender [56]. However, according to research on the development of children's drawing skills [56,57], students participating in the present study would be able to realistically describe the conservation value of the landscape.

There may also be some difficulties in using drawings as research data. According to Barraza [18], Backett-Milburn and McKie [58], Horstman [59] and others, the main drawback is the difficulty in analysing the drawings of young people in particular. In the present study, the drawings of the Mexican students were accompanied by explanatory texts which were used as additional support for the interpretations made on the basis of the drawings. Thus, versatile sources were used to understand the drawers' meaning-making [60] and to validate the interpretations (triangulation). To increase the reliability of the analyses and interpretations, two researchers analysed the drawings independently using mutually agreed upon criteria.

### 3.4. Study Method and Analyses

The data was categorized quantitatively and analysed using qualitative methods. First, to get an overview of the landscapes students deemed worthy of conserving, the drawings and texts of all students (N = 655) were reviewed in general. Drawings were excluded if they were drawn on colored paper, contained text in Spanish or did not include the participant's age or gender. Thus, a total of 440 drawings remained for analysis (n = 161 by the 1st and 2nd graders; n = 127 by the 3rd and 4th graders; and n = 152 by 5th and 6th graders). About half of the drawings were drawn by girls (49%, n = 217). While the focus was on the drawings, the texts related to the drawings were only used in this analysis phase to support the interpretation. A chi-square test of independence was carried out to test if there was a significant difference in the frequency of the landscape types drawn by girls and boys.

Second, the drawings were classified into three main categories: nature landscapes, built environments, and social landscapes [23]. This categorization has also been used in previous studies [20–22].

Third, the texts of the 5th and 6th graders were analysed in more detail in order to determine what they conveyed about students' environmental values and what they would tell us about students' environmental relationships. The focus was on these grade levels because their texts were longer and clearer than those of the younger students. Further, it was assumed that the older pupils would have a wider vocabulary, and that they could better describe and present their thoughts compared to the younger students.

### 3.4.1. Analysis of the Landscape Drawings of the Students

The first research question focused on what kind of landscapes Mexican students wanted to conserve. First the drawings were divided into three main landscape categories [23] using deductive

content analysis [61]. A drawing containing a nature landscape without man or signs of human activity was classified as a nature landscape. If the drawing portrayed signs of human activity, such as a road, a bridge or a building, it was classified as a built environment. Landscapes classified as built environments could still include natural elements such as the sun and trees. Finally, if a drawing contained man, it was classified as a social landscape. For example, a drawing depicting a girl in a swing by a small cabin in the woods was classified as a social landscape, although it also contained elements of a nature landscape and of the built environment. Social landscapes represent Willamo's [29] view of the relationship with nature, in which man is seen as one with the environment and part of a landscape worthy of conservation.

Thereafter, the drawings were analysed by gender and grade level. Next, for the purpose of taking a closer look at what was occurring in each landscape type, the landscapes were further examined through inductive content-based analysis [61]. The contents of the drawings were described, and their main features were highlighted. For example, in the case of nature landscapes, it was examined whether the drawings included forest, water or beaches, and, in the case of built landscapes, it was noted which kinds of elements were depicted. The elements in the drawings were not quantified. Willamo's [29] work on relationships with nature was used to examine the landscapes worth conserving. The aim was to investigate whether the students placed human beings in the environment or not.

### 3.4.2. Analysis of the Texts of the 5th and 6th Graders

The second research question focused on what environmental values were conveyed by the 5th and 6th graders' texts and what these values revealed about the students' environmental relationship. A total of 152 texts were analysed through a data-driven manner. At the end of the analyses, the environmental values that emerged from the data were reflected to the environmental values created by Kellert [30]. In terms of ethical consideration, the students´ environmental relationship was discussed from the perspective of the individual, species, ecosystem and global levels based on Littunen and Lähde [28]. The theories did not form the starting point of the analyses and they did not direct the analyses of the written text. In addition, the results were compared with those of Loughland et al. [2].

First, the students' texts were read through. Thereafter, the texts were analysed using inductive content-based analysis [57]. The unit of analysis was thematic, consisting of one or more sentences expressing environmental issues or a student's environmental relationship [62]. The environmental values expressed by the students that corresponded with themes of environmental issues (see Section 2.5) were written down. Two researchers read through the texts one by one and recorded the themes that occurred in the material. The researchers performed the final thematic analysis together. The final decisions were made through discussions, which continued until consensus was reached and clear arguments were found. As such decisions always include elements of subjective interpretation, group discussions about each text were therefore essential. Researcher triangulation was an essential part of the process. Our research group consisted of experts from biology and geography education, environmental education, sustainable development education, educational sciences and language education, and two of the authors were experienced teacher educators and researchers.

Of the students' 152 texts, themes could be identified in 111 of them. The texts, which included only a list of what the landscape contained, were left out of the thematic analysis. Also, any landscape descriptions that contained contradictions were not included in the analysis. Thus, 41 texts were omitted from the analysis. Of the texts analysed, 63 were written by the girls and 48 by the boys.

From the data analysis, six different themes that described the students' environmental values were determined. After these themes were identified, new themes did not emerge, so the material was considered saturated [62]. Expressions describing each theme were categorized. For example, the texts expressing responsibility for nature were categorized under the theme of responsibility. There could be several themes in one text, and each theme was analysed separately. Thereafter, occurrences of a theme were counted, as well as the proportional percentages for the girls and the boys. The environmental values identified with thematical analysis were used to reflect on what environmental values could tell

us about the environmental relationship of the students. Subsequently, the theoretical background of the study was used to support the analysis. The aim was to study how the students' environmental values related to the environmental values created by Kellert [7,8,30]. In addition, the qualities of the students' environmental relationships were clarified by comparing them to the qualities identified in the studies of Littunen and Lähde [28] and Loughland et al. [2]. Two members of the research team first independently conducted the categorization and the subsequent analysis regarding students' texts in order to ensure the reliability of the process. The analysis process was dialogical in nature. The discussion continued until consensus was reached and clear arguments were found.

## 4. Results

In this section, the drawings and texts, according to the main landscape types and the school grade groups, and the differences in values and environmental relationships between genders are presented.

### 4.1. The Landscapes the Mexican Students Wanted to Conserve

In the Mexican students' minds, all of the three main landscape types—nature, social and built—were worth conserving (Table 1). The amount of drawn nature landscapes increased, and the social landscapes decreased, with age. A chi-square test of independence showed that there was no significant association between gender and the type of landscape drawn, $X^2(2, N = 440) = 2.65, p = 27$. The drawings were carefully made, and the descriptions attached to them clearly written. The students depicted a variety of different landscapes both from their immediate surroundings and from outside their homeland. Multiple areas and continents were mentioned beyond Mexico, including North America, Australia and Africa. In addition, the students valued both of the polar regions as worthy sites of conservation. Europe and Asia were not mentioned in any of the descriptions. The drawings were mostly done with coloured pencils, but there were also a few works made with crayons. The students drew primarily existing realistic landscapes and in nearly all landscapes—whether built or social ones—nature (elements) were present underlining the meaning and importance of nature to the students. A few of the drawings depicted mermaids or other fairy-tale creatures.

**Table 1.** The drawings (n = 440) classified into the three main landscape types according to the gender.

|  | Nature Landscape | Social Landscape | Built Landscape |
|---|---|---|---|
| Girls (n = 217) | 138 | 48 | 31 |
| Boys (n = 223) | 125 | 58 | 40 |
| Total (n = 440) | 263 (60%) | 106 (24%) | 71 (16%) |

### 4.1.1. Nature Landscape

The landscapes which were comprised of natural elements only (Figure 1) were classified as nature landscapes. In these drawings, there were no people or traces of human activity. The majority of the students' drawings were nature landscapes (60%). The drawings depicted nature in various ways and included elements, such as mountains, aquatic environments, forests, glaciers, deserts and a wide variety of animals. Mountains, in particular, often appeared in the drawings, and some of the local mountains were named, most likely because the students were familiar with them. Beaches in the drawings were often named, such as Cancún Beach, which is the best-known beach in Mexico. Typical water environments were waterfalls, streams and rivers. The animals in the drawings were penguins, polar bears, whales, various fishes and domestic animals. The nature landscape was depicted positively and considered very important. Bright colours were often used in these drawings, which could be interpreted as expressing a positive relationship with nature.

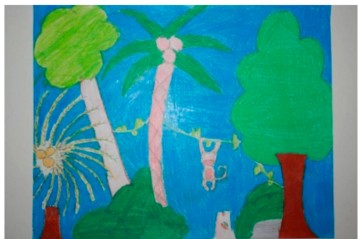 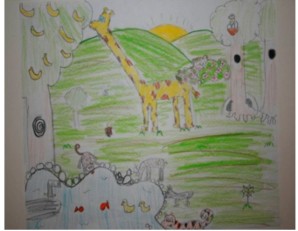 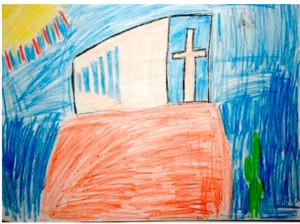

**Figure 1.** Nature landscape drawn by a 10-year-old girl (**left**), a social landscape drawn by a 10-year-old girl (the girl with long hair behind the flowers is looking at what is going on in the environment) and a built landscape drawn by a 12-year-old boy (**right**).

Illustrating a pure nature landscape was more popular among the older students, 3rd—-6th graders (about 70%), than among the 1st–2nd graders (43%) (Figure 2). This indicates that older students may be more appreciative of undisturbed nature than their younger schoolmates.

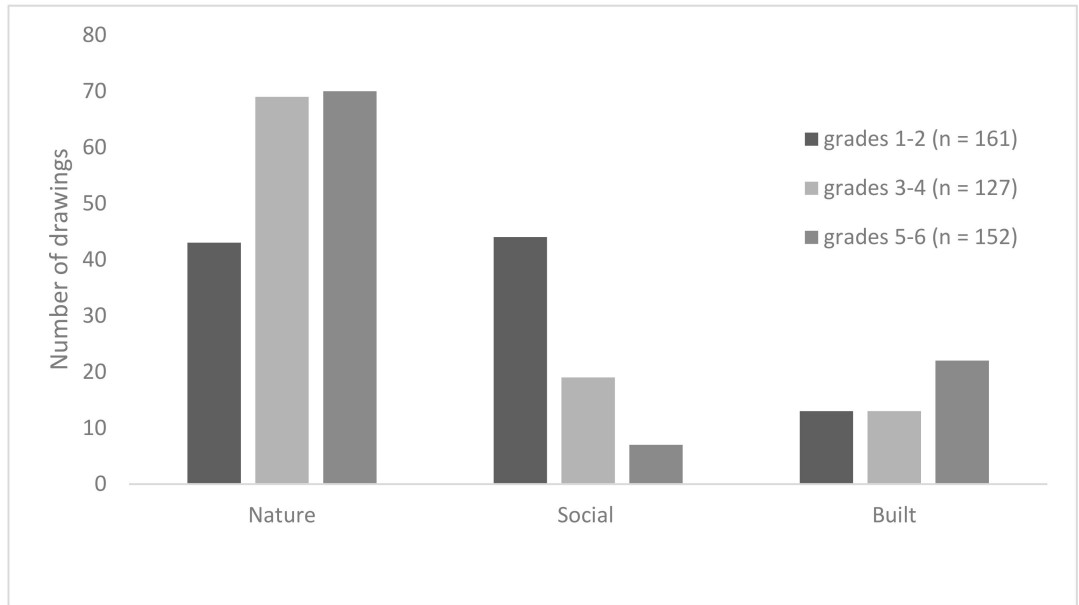

**Figure 2.** The drawings classified on the basis of the school grade groups and the three main landscape types.

### 4.1.2. Social Landscape

Those landscape drawings in which humans were present were classified as social landscapes (Figure 1). This was the second most popular theme after the nature landscape. Social landscapes were pictured in total of 106 drawings, making up 24% of the material. Most students drew the social landscape mixed with both the built and nature landscapes. These mixed-theme drawings made up 79% (n = 84) of the social landscape images. The social landscape, together with the nature landscape, was depicted in a total of 19 drawings (18%). There were only two drawings depicting a social landscape combined with a built landscape and one drawing with people as the sole element.

The social landscapes often illustrated different leisure activities, such as people hiking in the wild, paddling, fishing, downhill skiing, surfing on a surfboard or camping in the wild. A social landscape was portrayed in 44% of the drawings made by the 1st and 2nd graders, which was much more often those by the older students. Of the older students, 19% of the 3rd–4th graders and 7% of the 5th–6th graders drew a social landscape. In a few of their drawings, people were pictured beside dams and polluted water pipes.

### 4.1.3. Built Landscape

Drawings which showed traces of human activity but not people themselves were classified as built landscapes (Figure 1). There were 71 drawings of this kind, 16% in total. Most (n = 67) of the drawings which had built landscapes also had nature depicted in them (Figure 1). Illustrating a built landscape was slightly more popular among the 5th–6th graders than in the lower grades (Figure 1). About a fifth (22%) of the 5th–6th graders drew the built landscape as worth conserving, while only 13% of the 3rd–4th graders and 13% of the 1st–2nd graders valued the built landscape.

The built landscape was often drawn with a variety of different signs of human activity, such as cars, bridges, huts, parks and houses. Some of the 1st–2nd graders also drew other vehicles, such as airplanes and submarines. In the 3rd–4th graders' drawings, there also were buildings, fences and a pyramid. One of the most special was the drawing of the Mayan Temple Chichén Itzá, an archaeological site in Mexico. In their drawings, the 5th–6th graders depicted, for example, a church (Figure 1), parks with swings, a Christmas tree and an igloo.

The theme of littering was apparent with all groups. The youngest students drew pictures depicting issues related to recycling, such as recycling points or recyclable bottles and cans. The 3rd–4th graders likewise pictured bottles and cans representing littering in the landscape and environmental pollution. In their drawings, people who littered were often shown. The same theme emerged with the 5th–6th grade students. These students drew trash, signs prohibiting littering and a plastic bag in water causing harm to the fish.

There were no statistical differences in the main types of the landscapes drawn by the girls (n = 217) and boys (n = 223). The boys drew slightly more built and social landscapes (56%, n = 40 and 55%, n = 58, respectively) than the girls (44%, n = 31 and 45%, n = 48, respectively). Likewise, the girls sketched slightly more wildlife (52%, n = 138) than the boys (48%, n = 125).

### 4.2. The Environmental Values and Environmental Relationships of the 5th–6th Graders

The student's written descriptions of their environmental values depicted diverse relationships with nature and personal experiences related to landscapes. Multiple themes representing these relationships were identified in the data. These themes revealed their concern and affection for nature, and their texts often described causality and appreciation of aesthetics (Figure 3).

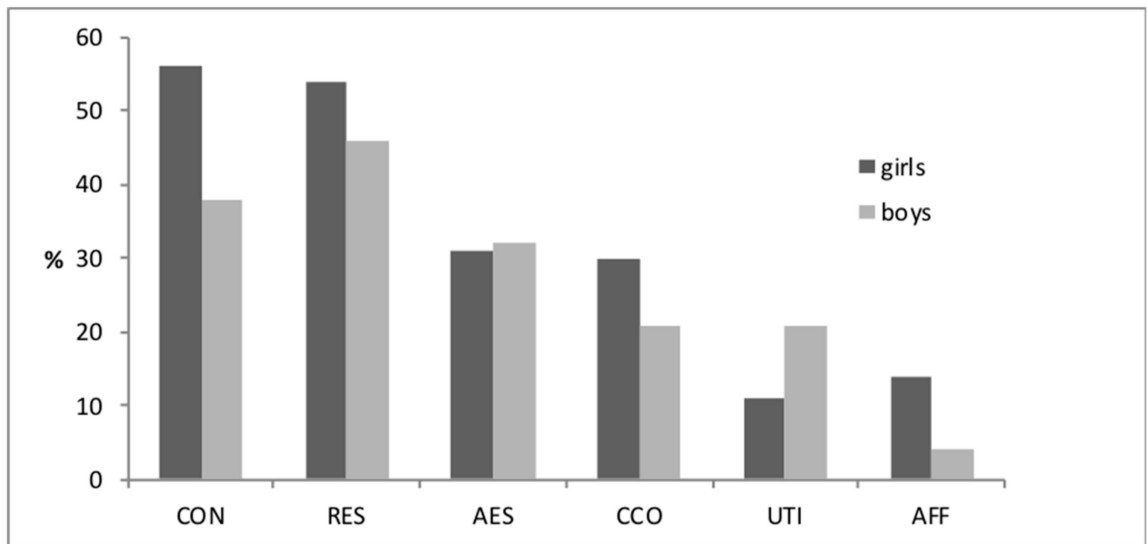

**Figure 3.** The environmental relationship of girls (n = 63) and boys (n = 48) with the landscapes they wanted to conserve. The percentage of respondents belonging to each thematic group is given. CON = concern; RES = responsibility; AES = aesthetic values, CCO = cause and consequences; UTI = utilitarian values; and AFF = affection for the landscapes.

In this section, thematic groups are used to represent the students' environmental values and environmental relationships. Themes are presented in order of frequency, so that the theme that occurred most often is shown first and the theme that was the least frequently mentioned is shown last. There were 124 themes mentioned in the girl's descriptions, approximately two themes per description. The boys had written about 77 different themes, having approximately 1.6 themes per description. Alongside the themes introduced, there are examples of the descriptions in which the theme was represented. Furthermore, in the theme analysis, attention was also paid to the possible differences between genders and the set of values concerning the landscape and environmental relationships of the Mexican students.

4.2.1. Concern for the Landscape

The assignment was to draw a landscape which the students found worthy of conserving. In addition, the students were asked to write a description of what they had drawn. Descriptions in which the students expressed worry or anxiety about environment, or about things threatening the landscape or animal species, were classified as the theme of concern. The way the assignment was given may have affected the frequency of the expression of worry, because one of the most recurring themes was worry. Worry was brought up in 53 descriptions. Expressing worry indicates that the students have developed the ability to care about the landscape and that the landscape was experienced as being important. Expressing worry points out that the students at this age are already aware of environmental issues. Also, the girls (n = 35) wrote about worry more often than the boys (n = 18).

The expressed worry was focused likewise on the state of the landscape, as well as on animals. In some of the descriptions, worry was also focused on people, but people themselves were generally felt to be the main cause of the worry. In these descriptions, people were seen as a threat to nature.

The juxtaposition of good and evil occurred in the descriptions. Two different landscapes were described, one landscape including conceptions of the unpolluted landscape and another one describing pollution threatening the landscape. The students showed worry about the declining state of the world and were worried that the bad version would be realized. One girl expressed this worry:

> 'I draw this picture that is divided in two parts the good, ecological part and the other part contaminated and polluted. We look like the bad part all contaminated and pollution everywhere. We should be the good ecological part. Let's change the world'. (Girl 356)

The students were worried about the state of the world, and they were also aware of current environmental problems. They presented a wide range of concerns, such as pollution, extinction, deforestation and climate change. In connection with pollution, they mentioned water pollution, an oil spill in the Gulf of Mexico and littering. The oil spill also caused concern about possible harm to animals and the landscape. Factories, excessive use of vehicles such as cars, and high energy consumption were also mentioned in connection with pollution.

> There's an oil spill in the Gulf of Mexico, and it reached the shore line. It had been the worst oil spill in the U.S. history. It surpassed the damage of the Exxon Valdes that spilled 11 million gallons and this oil spilled more than 11 million. It's important to be removed because many animals had been damaged and it's a lot of work for the people to green (clean) them and it's too many that it already reached the shoreline'. (Girl 471)

The previous description was classified as pertaining to the theme of worry, as the girl expressed worry about the well-being of the animals in the Gulf of Mexico and about the oil reaching the shoreline. Animals were mentioned in almost all of the descriptions of worry. This indicates that students are especially worried about animals and are prepared to act on behalf of nature and the landscape in order to improve the conditions of the animals. Extinction was often mentioned, as well as destroying the habitat of animals, for example, by felling a tree they live in. There was also worry about killing wild animals or catching them alive and then selling them. The students named several species of animals

in danger of extinction. A special concern was for big mammals. For the most part, the students were able to name animals in danger of extinction, such as polar bears and pandas. One girl wrote the following about her worry for the koalas and their habitat:

> *'I chose to save this landscape because of the koalas. Koalas are small animals that are in trees. They are little and fragile and they need our help. This little guy is so cute and endangered (endangered). I love to help them and keep them habitat'. (Girl 417)*

The polar icefields were often mentioned. Both the girls and the boys wrote and drew many depictions of both the North and the South Poles and Antarctica. The boys mentioned mostly Arctic and Antarctic animals, but the girls also noted domestic animals they were worried about. The students wrote about the warming of the climate and the melting of the ice. The melting of the icebergs was connected to the disappearance of the habitats of the birds in the Arctic and Antarctic regions. Polar bears, seals and penguins were mentioned in this instance. It would be interesting to know why these topics were present in so many descriptions. Perhaps these topics had been discussed in school recently.

> *'I want to preserve the north pole. Because of pollution the north pole is melting and animals are dying. Animals in there are being in danger. And it is all our fault. Because of using the energy in excess the poles are melting and the global warming is increasing . . . ' (Girl 477)*

In some texts, the students also expressed their worry about the bad condition of the landscape and the effect of human's activities on future generations.

### 4.2.2. Responsibility for the Landscape

Concern or worry was often accompanied by the mention of responsibility in the students' texts. This theme was chosen when human beings were seen as responsible for the described environmental disaster, or strong reasons for conserving certain things were given. Awareness of human's responsibility for nature was apparent in the students' writings. The theme of responsibility was also included in the students' writings when they gave advice on how to help the landscape and animals, such as 'Let's recycle!' or 'Don't throw trash!'. Also, writings about how human beings in general should act to conserve the landscape were observed. Many students wrote 'we', for example, 'We have to save the earth!' (Girl 421), thus underlining their understanding of our common responsibility for nature.

There were 66 remarks under the theme of responsibility, 34 of them written by girls and 22 by boys. There was no statistical difference between the genders: 54% of the girls and 46% of the boys brought out this theme in their writings. Both girls and boys named things humans should take responsibility for and protect. The girls wrote about the conservation of the sea or a distinct landscape for the benefit of future generations, such as their own children. The boys' texts reflected human's responsibility for even more distant landscapes and animals. The students' message was that our current lifestyle leads to the pollution of our surrounding landscape, and, as a result of our choices, the climate is warming up, which may have severe long-term effects.

> *'My landscape is a river and a water falling the mountain. And I made it because I want to save fresh water'. (Boy 440)*

> *'It represents all the nature around. Let's keep it like this! Let's save trees and the flora and fauna. There are also many trees and fresh water'. (Girl 453)*

In their writing, the students stated that they needed to take responsibility (I will, I have to), they needed to join the group that should take responsibility (we should, we have to) or that responsibility needed to be taken by human beings (people should).

### 4.2.3. The Aesthetic Values of the Landscape

The aesthetic values of the landscape, that reflect its beauty and importance as a cause for conserving the landscape, were mentioned in 35 students' texts. Both the girls (32%) and boys (31%) valued the aesthetic nature of the landscape. They wrote about the beauty of the nature and of the landscape. In almost all of the texts that mentioned this theme, the aesthetic values were connected to the landscape.

The students often mentioned mountains, water environments, forests and deserts as beautiful places. In particular, such water environments as seas, ponds, rivers and waterfalls were mentioned. Many students wrote about distinct nature landscapes in Mexico. Also, many Mexican mountains were drawn by the students.

> *'My beautiful landscape is found in Mexico. It is Cancun, Quintana Roo in Mexico ... It is very clean without oil or many other trash ... It is a beautiful sunset on it. I chose to save that landscape because it is very beautiful'. (Girl 410)*

The students wrote that there was no place for litter or pollution in their valued landscape—on the contrary, they appreciated the greenness of nature. Valuing a clean landscape was brought up in eight of the writings.

> *'It represents the beautiful mountains without houses in it and the air is fresh and with a beautiful sunshine.' (Girl 368)*

There were only two texts endorsing a built landscape as a beautiful landscape. An example is where a boy wanted to bring out the beauty of a building sitting in the landscape.

> *'This is a church in Arizona that is in a top of mountain. I wanna conserve this because is so beautiful.' (Boy 508)*

### 4.2.4. Relation between Cause and Consequences

In the writings classified as relating to the theme of 'relation between cause and the consequences', the students often externalized themselves and talked about the mechanisms threatening nature or of those already realized as a matter of cause and effect. In the following text, a girl discussed climate change causing the melting of ice fields which causes a threat to the environment of polar bears:

> *'Polar bears in danger of extinction. They live on the ice, and ice is melting because of global warming'. (Girl 360)*

This example also reflects the connection between the protection of the landscape and the conservation of species. The writer does not express her emotional affection to the landscape; rather, she describes the landscape as a bystander.

The students' writings were not presented scientifically using the terminology of the natural sciences. In these texts, a simplified view of the processes taking place in the landscape was expressed. The texts often discussed what happens and its consequences. Many noted the lack of an explanation for why something was happening.

The relation between cause and effect was mentioned 29 times, and more often in the girls' (30%) than in the boys' (21%) texts. In the following example, a girl noted how she wants to conserve all the lakes in the world to prevent the animals dying from the litter people throw. The text was classified as belonging to both the 'cause and consequences' and 'responsibility' themes. Responsibility is shown by her wish to conserve all the lakes, and cause and the effect by her description of the events, where throwing litter in the water causes the death of the animals.

> *'I want to save lakes all over the world. With animals inside, because people throw trash in lakes so that makes animals to die ... ' (Girl 463)*

### 4.2.5. Utilitarian Values of the Landscape

This theme describes situations where the writer, or people in general, benefit from the thing mentioned in the text. In the writer's mind, its being useful to man would be a reason to conserve this landscape. The utilitarian point of view was brought up more often by the boys (21%) than by the girls (11%), totaling 17, which was only a fraction (9%) of all the themes (n = 201).

One of the examples of the benefits was a comfortable place in which one could quiet down or play sports. In this study, a mental advantage was also included in the benefits gained, in addition to material benefits, as one boy wrote:

*'Hills because you can do a lot of things there'. (Boy 355)*

Some of the texts stressed the importance of nature in maintaining life, and the preservation of trees was argued by some to be essential for the production of oxygen. Also, securing a supply of pure water was mentioned, since people need it to survive. For example, one student wrote:

*'I would like to save water. Because without water we can't live'. (Boy 491)*

Some of the texts emphasizing benefits described a chain of events or a natural process, for example, where a girl wrote about the importance of trees for the life of human beings and animals. In many of these texts, both 'cause and the consequences' and 'utilitarian' themes were present.

*'I want to save a tree because trees are very important in our lives, they give us oxygen, beauty, sometimes they even give us food. Trees are not only important for us they are important to animals to . . . ' (Girl 496)*

*'I want to save trees to give us oxygen to breathe'. (Boy 499)*

### 4.2.6. Affection for the Landscape

Texts showing the students' affection for the landscape they had drawn were included in this theme. Personal affection for the environment was found in 11 texts only, nine of those written by girls (14%) and two by boys (4%). In most of the texts, the students expressed their personal affection with the word 'love', which showed their deep attachment to the landscape.

The students often explained what was so attractive in the landscape they had drawn. They accurately described what kind of effects this important landscape had on them. They wrote that landscapes can calm and relax you and that, while looking at them, all your troubles will be forgotten.

Affection was mostly felt for a landscape in the nearby surroundings of a student. Here is an example, where a boy describes a river, which runs by his friend's grandma's house.

*'I drew that because I went with [my friend] and I loved that place. We went to a river and we play and swim on the water. And we dive. That place was beside Enrique's grandmothers house'. (Boy 458)*

In addition to the nearby landscapes, affection was mentioned in regard to foreign landscapes, such as Australia and Hawaii. These may have become important to the students because they were places where their families spent their holidays. However, most of the places mentioned were in Mexico.

Affection for animals was often expressed. The students described the animals they had drawn into the landscape and brought up the importance and beauty of both domestic and wild animals. In their texts, the students often showed affection for a landscape because there were animals in it.

*' . . . I love this great big landscape. I chose to save this landscape because of the koalas . . . ' (Girl 417)*

## 5. Discussion

This qualitative case study focused on 7–12-year old Mexican students' drawings of landscapes they deemed worth conserving to determine the students' environmental relationship and environmental values. The participants were students of a private immersive English language school. The drawings were studied using deductive and inductive content analyses.

The first research question concerned *the landscape types* presented in the drawings of the Mexican students. The pictures were drawn according to instructions. The content was clearly drawn, and the attached descriptions were well written. The students drew all three main landscape types: nature, built and social. Built and social landscapes were often connected to the nature landscape [8]. Younger students (1st–2nd graders) drew more social landscapes than the older ones. The human beings in these social landscapes reflected the young students' own experiences with the nature, as was also the case in studies of Finnish and Swedish students [20,25]. These social landscapes indicated that students have a close relationship with the places drawn. This result was in line with one study of Australian adolescents who thought of nature as a place having meaning to them, especially regarding their psycho-emotional well-being [63]. The older students, the 3rd–4th and 5–6th graders, drew more nature landscapes than the younger ones. These results were also in line with previous studies [20–23,25].

Students drew their immediate surroundings, as well as foreign environments, for example, the North Pole. The landscapes the students wanted to conserve reflected positive things, such as landscapes with relaxing places or with biodiversity. The older students drew more remote places, and this result indicated that the students had environmental knowledge of various places on the globe. They also were able to connect the drawn places to global environmental issues. Animals were often present in the drawings and in the texts, and many of the texts encouraged humans to protect animals and their habitats. This differed in part from earlier studies [20,24] in which animals were rarely drawn in the landscapes but was in congruence with Gambian and Kenyan students' drawings, where animals were often present [64].

The environmental values conveyed by the 5th and 6th graders' texts related to the second research question. The students expressed their personal values, feelings, impressions and opinions, and they also described environmental threats. The same expressions were found in Finnish and Swedish students' drawings and texts [64]. However, for the first time, in this study, the environmental values were also categorized, and six different environmental themes were found. The most often recurring theme was concern, although some students also expressed anxiety, possibly reflecting general discussion of environmental disasters at that time or what they have learned at school. These students had learned about the Exxon Valdes oil spill in their reading class highlights. The students showed concern for the animals, environment, environmental problems and future generations. Being concerned and expressing worry about the environment showed that these issues are highly important to the students, in keeping with Kellert's humanistic approach [8,30].

More specifically, students were concerned about the state of the environment and the animals. The concern for the fate of animals evidenced in the material was not surprising, as it is typical for children and adolescents of this age to be particularly interested in the living creatures of nature. Students can emotionally attach themselves to animals very easily [8]. Concern for animals also emerged in the studies of Palmer [65] and Bonnett and Williams [43]. Large mammals were considered important in the environment, and the children indicated that they were emotionally attached to them. These results are in line with earlier studies [66–68] where children in their drawings focused on large mammals or were concerned about remote species, such as in Alerby's [19] study. Large mammals were most frequently mentioned in respect to distant species whose habitats are endangered. Attachment to large and endangered animals may be due, for example, to the presence of large mammals and the issue of their extinction in nature programs and documentaries. In these studies, the students wrote a lot about animals, and their lives were seen as valuable to humans. Similar results were obtained by Bonnett and Williams [37].

The Mexican students highlighted various environmental issues in their texts, such as global warming, pollution and the felling of trees. It is typical of 5th–6th graders to pay attention to the abnormality of natural phenomena [30]. Many previous research studies have shown that children can be worried about the loss of the habitats of animals [18,19,37]. In Barraza's [18] study of British and Mexican children's perceptions and concerns about the environment, the students also highlighted cause and consequences relationships, which described, for example, global warming and the loss of habitat for polar animals. Such relationships were also found in this study. The oil spill in the Gulf of Mexico received some mention. The students were aware of the damage it had caused and were able to provide very accurate information about the disaster. They were particularly concerned about animals suffering as a result of the oil spill. The students were able to discuss the oil pollution damage at school, but information was also available in local papers and through television news. In fact, a global examination of environmental problems indicated that the environmental relationship and ethical consideration of children are already at a high level [24].

Concern and *responsibility* for the environment were often mentioned together. The Mexican students were worried about their environment, and they were ready to act to protect nature. They also encouraged others to take environmentally friendly actions. This showed that they have an idea about humans' responsibility regarding nature. The same awareness about responsibility was found in the studies by Alerby [19] and by Loughland et al. [2], where children knew the responsibility that humans have toward the environment. The students mentioned many global environmental problems, such as global warming, pollution and deforestation. According to Kellert [8], it is typical for students at this age to notice abnormalities in nature.

*Aesthetic values and affection* for the environment were mostly shown in the texts concerning the Mexican landscapes, especially the mountains. This result supports the observation of Eagles and Muffit [53] that, among 12 to 14-year old students, humanistic values are popular. According to White [17], children typically can be very emotionally attached to a familiar and comfortable environment when allowed to establish a genuine relationship with nature. The texts attached to the drawings also revealed the existence of the relationship between *cause and consequences* in the landscapes and *utilitarian* values. According to White [69], alienation from nature can transform children's mindsets to see nature as being subordinate to human. However, the results of this study differ from White's results. All the students showed a positive or neutral relationship with the environment, and nature was not seen as being subordinate to human.

The girls described their *relationship* with the environment in more varied ways than boys. They wrote longer descriptions of the landscapes worth preserving, and that is why their texts contained more themes than the texts of the boys. This may be because girls' writing skills are better than those of boys [66]. The girls raised concerns especially about the environment. In a study by Eagles and Demare [5], girls were found to have a higher moral attitude toward nature than boys. It is difficult to gauge the level of morality in this study, but it is worth noting that the girls' responses emphasized a high degree of moral responsibility.

Kellerts [7,8,30] humanistic and moralistic attitudes among students were reflected via concern, responsibility and affection. And, finally, the student-generated material emphasized the nature-centred environmental relationship of the students.

## 6. Conclusions and Implications

This study provides an example of how teachers can determine crucial information about students' interests and environmental values. More attention should be paid to the meaning that students attach to their everyday environments. This is important because schools play a key role in developing students' environmental awareness and positive environmental values. It was noteworthy that Kellert's [8] environmental values were also found in this study. Humanistic, aesthetic, moralistic and utilitarian values are closely connected to the themes of responsibility, worry, aesthetics and benefits. This reveals something about the universality of environmental values found in all kinds of cultures.

Recognizing the potential of culturally oriented education could strengthen the ties between content knowledge education and sustainability education.

As students showed concern about preserving nature, teachers should discuss with them about different ways to take action and make changes. Concerning, e.g., the environmental disasters, discussions are important in order to reduce students´ anxiety which has emerged, for example, because of the climate change issue. Based on these results, humans´ well-being in connection to the landscape and environment should be studied further in the future. Another future approach to be studied would be the effect of climate change: it has increasingly become an issue of global concern, and as children around the world have become involved in different types of activism, it would be important to see how this is reflected in children at the age of those participating in our study.

**Author Contributions:** Conceptualization, E.Y.-P., E.J. and G.R.-A.; Formal analysis, E.Y.-P.; Investigation, E.Y.-P.; Methodology, E.Y.-P.; Supervision, E.Y.-P.; Writing—original draft, E.Y.-P., E.J. and G.R.-A.; Writing—review & editing, E.Y.-P., E.J. and G.R.-A. All authors have read and agreed to the published version of the manuscript.

**Funding:** This research received no external funding.

**Acknowledgments:** The authors want to thank warmly teachers Outi Ojansuu and Maarit Tyynismaa for their precise preliminary analyses and the idea of the title.

**Conflicts of Interest:** The authors declare no conflicts of interest.

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
