# Peer review of "‘Nature Is Something We Can’t Replace’: Mexican Students’ Views of the Landscape They Want to Conserve"

_education, doi:10.3390/educsci10010013_

Round 1

Reviewer 1 Report

This is a well-written paper about how children perceive landscapes and how they express ideas about conserving landscapes. However, I find some  areas that need to be addressed:

What is the academic contribution of this paper? Some of the connections made are rather tenuous. We do not really understand children’s relationship to nature through this study, and yet that is an assertion. What does it tell us that children drew social, natural, and built environments? Children around the world inhabit these three spaces, and without greater context to children’s lives and experiences in the context of Monterrey, Mexico, there is not really much of academic interest that this study can tell us besides that they drew three landscape types and showed concern for the environment. So then, what is the academic contribution of this paper? Why connect landscape drawing analysis as a way to understand what children value about nature? The landscape drawing analysis seems focused on understanding children’s conceptions of landscape, but as the children’s concerns show, this is not really connected to their ideas of conservation that they reflect. Concerns about oil pollution or global warming are not concerns about landscapes, per se, but about degradation of the environment in general. What is the context of these children’s perceptions within Mexico? You do not give a contextualized framing of this research in the context of the city of Monterrey, or in Mexico more generally. And the children who are subjects of the study are in no way reflective of a cross-section of Mexican children more generally. It would appear by the authors’ references to other study sites that the hope is to generalize findings about landscape from other places. But to generalize, one needs to have a representative sample of a population, which this study does not do. From my perspective, it is trying to do two things, and I am not sure it is accomplishing either well. It is placing children’s experiences in the context of a geographical place, to understand perceptions of landscape. And yet that geographical place is not described or considered in the article. OR it is trying to generalize about how children around the world view and value the environment, in which case a different methodology would better support such generalizations. I think that either would make a fine contribution, but right now, I do not see how it is successfully doing either of these. Why draw so strongly from Kellert’s values typology? And then not use it in your analysis. The values typology is fine for an analysis tool, but the circular means of inductive analysis and then linking this back to the typology in the discussion makes no sense to me. Which again leads me to: what are the purposes of this research? What academic contribution are you really trying to make?

I have a number of specific questions/concerns below as well.

I was confused by the paragraph, page 1 that begins: "The Finnish word for landscape ‘maisema’". With all the references to European words. Isn't this a study about Mexican children? And therefore, wouldn't it be more appropriate to be discussion conceptions of the words they use in their own languages (Spanish and any indigenous)? This paragraph also seems like a detail in an otherwise conceptual framing section of the paper, and so it seems misplaced (in addition to confusing about why it is there at all).

At the bottom of page 1:

I find this statement problematic and somewhat inaccurate: “Childhood environments as well as cultural and environmental experiences create the basis of the environmental relationship [5].” The authors cite Eagles which is an article about environmental attitudes, not relationships. For relationships with the environment, it seems children’s place attachment literature and children’s geography would have much more to say, and it would seem that some factors are missing in this brief description.

I find the distinctions between values, attitudes, and relationships with the environment confusing and without much differentiation in the way they are currently described. In the current environmental movement, why focus on values and attitudes as separated from action and agency? We know children are inheriting many environmental problems, and their own perceptions of these problems are well documented in the academic literature. Focusing on values and their identification of concerns alone seems slightly out-moded in terms of informing environmental education.

Pg. 2, line 49: it is important to say that scholars ASSERT that children spend much less time in nature than in RECENT generations. We do not have very good baseline data about how children have actually spent time in nature. It is possible that they were outdoors, but not in nature, or outdoors, but not really interacting with nature, in many contexts of recent history. Attribute these ideas as concerns of children’s advocates and environmentalists, not as a given fact for which we do not have very good data. Or provide specific, contextual data that compares children in a setting both before and after. Or speak to increasing urbanization. Or in some way be more precise. And cite scholarly, peer-reviewed sources for evidence of this.

Explain why you are interested in these particular research aims? What will they contribute, what gaps in existing knowledge?

Pg. 2, when introducing the qualitative study. Explain why Mexican children. And contextualize with research that already is conducted in Mexico, such as Tuline Gulgonen and Yolanda Corona’s work of Mexican children and drawings in a wide range of contexts. Why Monterrey? Mexico is a country with great environmental and cultural diversity. How does Monterrey contribute to an understanding or generalization about Mexican children in general (which the study largely purports through its title and framing)? What are the authors’ familiarity with Monterrey and its context?

Pg. 2, why ages 7-12? Explain this age group in the context of childhood development and relationships with nature (e.g., David Sobel’s assertions about middle childhood and nature, or Chawla’s childhood place attachment theory for middle childhood).

I do not understand the research value of having students draw and then categorizing drawings into “natural,” “social,” or “built.” I understand the categories, but not how this contributes to an actual view of landscapes but rather places. And how do you account for the somewhat randomness of what children might decide to draw on a particular day? Why not ask children about a favorite place or a special outdoor place instead?

Pg. 6, Section 3.2. Much greater contextualization of the children’s experience and community is needed here. Why, beyond, convenience, is this a good sample to explore these questions with? Wouldn’t a stratified sample with children of lower incomes, who are much likely to have very different experiences outside or in nature, also be important? What is the goal in approaching students in the particular context that you did? What is the experience of the study participants? Where do they live (Monterrey is a very large city, so what is the context of their daily lives, and what is the extent of their experience traveling on vacations in Mexico or other places?)

Pg. 6, Section 3.3, the description of limitations of drawing analysis, and how that was addressed are good. Provide also the inter-rater reliability of the two researcher’s drawing analyses. Were both researchers fluent in Spanish in order to interpret the data? [Page 7, why couldn’t you include drawings that had Spanish annotations? Why were these students’ excluded?] Did the researchers have an understanding of the landscapes in and around Monterrey in Mexico to understand salience of details in the drawings?

Pg. 7, Line 292. This sentence does not make sense: “The data were categorized quantitatively and analysed using qualitative methods.” And if qualitative data were coded and counted, then provide the inter-rater reliability.

Pg. 7. So many drawings were excluded, throughout the description of the analysis, it starts to feel problematic. Why not collect a larger sample from those students who had the developmental ability to write? Or why not interview the students who were younger and could not annotate. There are so many ways data were excluded (language, writing ability, convenience sample of a single school in the first place). As I read on, it starts to feel that this is a very narrow glimpse of students, and really selecting for what is likely to be the most privileged and elite of an already elite group. So then this is a study that is mired in class, which is already a potentially large factor for children’s perspectives in Mexico. Which gets back to the research purposes, and if to generalize, then you need a different study sample.

Pg. 7, Line 330, the sentence just ends with a big context missing.

Pg. 8, I do not understand why the coding of the data was inductive if it was looking for Kellert’s values? Your explanation of how it was inductive, but not why you chose this approach rather than deductively coding for Kellert’s values and then if new categories appeared, adding them. From your analysis, it appears that actually no new values emerged, although you call them by different names. See below. I appreciate the triangulation of researcher analysis. Please also explain the researchers’ familiarity with Monterrey, Mexico.

Related to this, the thematic analysis feels confusing. “responsibility” would be considered “moralistic” value under Kellert’s typology, and so would “concern”. “affection” would be considered “humanistic”. Why did you not use the typology values that you wanted to explore directly for your analysis and then only if something actually does not fit into the typology would you create a new category? Based on the categories you present in Figure 3, all of these are actually part of his typology. And in fact, you come to this in your own discussion. I think the analysis needs to be rethought if it is to actually reflect the stated aims of the paper.

Finally, the end discusses that this article will contribute to teachers education about geography, but I fail to see how it will do that. The paper is not focused on educational approaches to engaging children.

Small details:

Pg. 2, line 45 and 89 (and throughout): change “man” to “people” or “humans”. “Man” is no longer an acceptable term to generalize about males and females.

Pg. 8, “thematic analysis”, not “thematical”

Reviewer 2 Report

The connections between cause and causation, environmental education, and so much more are truly wonderful to see in this article and I salute the authors on their excellent work.   

I only hope that given the importance of this article, that those studying university students and the general public will know about this study in the future, to inform their own studies.  Perhaps the authors could make a storymap or a web page to help educate others beyond those who will read this journal article.

The only thing I could suggest is making connections to studies that analyze drawings, and some possible items to be learned from such drawings, and what this study to contribute to such studies.  

Also, I am left wondering how much field work these students have had, and connections to Richard Louv's "Last Child in the Woods" research about environmental attitudes and immersive outdoor experiences.  Perhaps the authors could spend 1 sentence on this and speculate.

Have any of these students used any digital mapping, GPS, GIS, or remote sensing in the past?

The observations are quite wonderfully detailed but if there are space constraints, I wonder if the authors could and should reduce these in order to spend a few more sentences on the conclusions and wider implications; again because the study might be covering one school in Mexico, but it has much broader implications.  Also, I believe the readers will be interested in reading the authors' recommendations for future studies, and about environmental and science and geography education.  

Reviewer 3 Report

I like the idea to let children paint the landscapes they would like to conserve. In this way we get interesting insides into students’ images that are activated when they have to think about conservation of landscapes.

My main problem with this manuscript is the question why this research has been done. That means that the theoretical introduction does not outline questions which refer to a desideratum of research: Why are Mexican students analyzed? The focus on Mexican students suggests that a cultural comparison is the aim. But no theoretical considerations about possible cultural differences in imaging endangered landscapes are made. Another possible focus could be an in-depth analysis of environmental education in Mexico and its possible effects on students’ images of endangered landscapes. But such an approach lacks, too.

From earlier research it is known, that the paintings can be categorized according to “natural”, “built” and “social”. Therefore, the general finding that all three categories could be found within this population is very expectable. The differences between the ages could be of interest, relating to a psychological development perspective. But, if this would be the aim, an appropriate statistic and a much broader sampling would be required. Results from Swedish, Finish and Russian children exist (Yli-Panula, Persson, Jeronen, Eloranta and Pakula 2019): why not merging these data to analyze such a general development by a sound statistical analysis? An analysis of the landscapes by other research questions is done only in a superficial way with unclear criteria and without presented statistical data. Especially the question, how many and which students paint landscapes from Mexico or from distant places would be of interest. Accordingly, the analysis of the Mexican paintings doesn’t reveal much new insights beyond the results of the study of Yli-Panula, Persson, Jeronen, Eloranta and Pakula (2019).

The research about the environmental values underlying the paintings misses a clear question, too. Again, the findings, that Kellerts categories can be used, are quite expectable. If a deep analysis of students’ environmental values is the aim, other methods should be used (e.g. interviews or written tasks with questions that address the values more directly).  

The parallel analysis of the landscapes and the values interpreted from the paintings and the attached texts suggests that the relationship between these results would be analyzed: Do specific values and specific landscapes correlate? Unfortunately, this is not been done. In this way, it would make sense to analyze the environmental values of students by their paintings and attached texts.

In general: I think, this research project has gained fruitful data. But these data should be analyzed by more clear questions in a statistical way.

Some more specific remarks:

Obviously, the data were collected around 2010, when the oil spill in the Gulf of Mexico happened (deepwater horizon). But nowhere the time of data sampling is mentioned. The literature consists of work mostly published before 2011.

35-38 This part is quite different from the text before and after. It is not clear, why this – important – question is mentioned here. 247-249 Only two enumerations instead of three. In this way the two research questions are highlighted. 325-329 This part ends suddenly.

Study Method and Analyses: How many tandems of researchers analyzed the paintings and texts?

Reviewer 4 Report

[review text omitted: it was posted to a different submission]

Author Response

See the comments to the reviewer 3. 

These comments seemed to be downloaded twice.

Reviewer 5 Report

I found this manuscript to be interesting, given my limited knowledge of the use of illustrations for collecting data. The articulation of thought throughout the paper tends to be good although the use of simple sentences and ‘extra’ words in places is a bit of a concern. There is good use of the literature to build and support the case – nice job! Saying this, the author(s) might connect the literature in the Results/Students comments section in order to support some of the comments. This was attended to better in the Discussion portion of the paper.

Given the data was collected from a large group of higher socioeconomic status students, I wonder how this information would compare with students from lower socioeconomic status – one would assume that the status would probably allow for some travel, whether nationally or internationally while those without money may not have the same opportunities. Did this concern come up in the literature and if so, should you comment on this in your discussion section? And if it didn’t come up, perhaps commenting on it, further than a point in the paper about students may have been taught something recently thus the number of students who attended to a specific environment or issue.  

Perhaps putting the two research questions in the Abstract so readers know up front what the research is driving at … you did tell us what results you were looking for but the questions would aid the reader in specifically understanding what your study was all about.

My biggest concern of this paper is in the Methodology section. There was little information provided about who administered the tests to the various classrooms. At one point it was mentioned that a postgraduate student collected the data which left me wondering why of the 565 total drawings, only 440 drawings were considered? (“Such drawings were excluded which were drawn on coloured paper, had text in Spanish or did not include the participant’s age or gender”). I would have thought/hoped there was consistency in the handing out of materials, stated instructions (exact script followed for each class), same amount of time, same comments to each group, and so on. Without identical (or as close as possible) instructions one might question the validity of the results. Perhaps the author(s) can attend to this concern more specifically in the methodology section.  

Following are thoughts I had about specific lines in the manuscript (sentences or words in italics are the author(s):

Line 19 – 22 – “Concern or worry was often accompanied by the mention of responsibility in the students’ texts. It appears that the students were not familiar with scientific terminology and the natural processes and depicted the threats to nature in everyday language, nor did they consider their own activities in relation to the environment.” This is an awkward sentence and construes a couple of meanings.

Line 22 – 24 – “Knowing scientific concepts and processes is essential to developing an understanding of the environment as a physical and mental entirety.” Not quite sure what the author means – please provide clarification.

Lines 100 – 102 – “The landscape and the environment gain their significance through individual experiences, and observations [6,27] have shown how environmental relations and images in space and place are created and how they are shaped by knowledge, experience and thinking.” I’m assuming the student background experiences were not considered in your study – prior experiences could certainly account for how students would view/create various landscapes that they drew/wrote about. Perhaps flushing out the sentence would be beneficial.

Line 158 – “The environmental relationship is shaped by various factors over many years, and the basis of the environmental relationship is created in childhood [5].” What are examples of the factors? What is the span of childhood in years? Sentence might contradict itself

Line 195 – enquiryvs inquiry?

Line 247 - 1. The study questions were the following:- Why the number before the statement about the study quesions? I don’t believe a number is required before the statement about the study questions.

Line 261 – “Data were collected….“ – were versus was. Same comment for line 292

Line 294 – “Such drawings were excluded which were drawn on coloured paper, had text in Spanish or did not include the participant’s age or gender. Thus, a total of 440 drawings remained for analysis ….” 115 participants is a significant number removed from the research data

Line 316 – add ‘as’ prior to “…. a road, a bridge …..

Line 340/341 – “The unit of analysis was an idea of one or more sentences that expressed environmental issues or the environmental relationship [58].” I’m not sure what this sentence means

Lines 476/477 – this is a very important point! “The way the assignment was given may have affected the frequency of the expression of worry because one of the most recurring themes was worry.” How was the task administered and why would the way it was given affect an emotion of worry? This point should be addressed in the methodology section.

Line 663 – “The landscapes the students wanted to conserve reflected positive things.” Give an example or two of the positive things.

Lines 650 and 670 - In the Discussion portion of the paper you might reintroduce the two research question. I believe they are only specifically stated once in the paper

Line 674 – “The most often recurring was worry.” This is an awkward sentence. I’m wondering if the term ‘worry’ was used by the students or the researchers constructed this term for the paper? Saying this, I wonder if this ‘worry’ speaks to how we teach environment/environmental education to students. And does this speak to how anxious children are about the ‘state of the world?’ This too is an issue. Is ‘worry’ and ‘concern’ the same? You mentioned ‘worry’ and then immediately went onto ‘concern’, thus my question. If they are the same (to you as the author(s) then you need to make a note about this). What are the other ‘six different environmental themes’ that were found?

Line 724 – “The texts of the girls were longer than those of the boys.” Unnecessary sentence given the prior thought that girls wrote longer descriptions

Line 864 – font size in this reference is different

References – why are some years in bold while others not?

Round 2

Reviewer 3 Report

The aims of this study and its role within a larger research project are presented much more clearly than before. In this way the specific results of the Mexican part of the bigger project are made accessible. The interpretation of the results is well done.

Author Response

Thank you for your time to review our manuscript the second time. The asked changes have been made. Please see the manuscript with changes.